# Absolute Configurations and Chitinase Inhibitions of Quinazoline-Containing Diketopiperazines from the Marine-Derived Fungus *Penicillium polonicum*

**DOI:** 10.3390/md18090479

**Published:** 2020-09-21

**Authors:** Xing-Chen Guo, Ya-Hui Zhang, Wen-Bin Gao, Li Pan, Hua-Jie Zhu, Fei Cao

**Affiliations:** 1College of Pharmaceutical Sciences, Institute of Life Science and Green Development, Key Laboratory of Medicinal Chemistry and Molecular Diagnosis of Ministry of Education, Hebei University, Baoding 071002, China; guoxingchen92@163.com (X.-C.G.); 15689932652@163.com (Y.-H.Z.); 2Coollege of Life Science, Cangzhou Normal University, Cangzhou 061001, China; wenbinxing@yeah.net; 3State Key Laboratory of NBC Protection for Civilian, Beijing 102205, China; bk6180b@163.com

**Keywords:** marine-derived fungus, *Penicillium polonicum*, quinazoline, diketopiperazine, bioactivity

## Abstract

Three new quinazoline-containing diketopiperazines, polonimides A–C (**1**–**3**), along with four analogues (**4**–**7**), were obtained from the marine-derived fungus *Penicillium polonicum*. Among them, **2** and **4**, **3** and **5** were epimers, respectively, resulting the difficulty in the determination of their configurations. The configurations of **1**–**3** were determined by 1D nuclear overhauser effect (NOE), Marfey and electron circular dichroism (ECD) methods. Nuclear magnetic resonance (NMR) calculation with the combination of DP4plus probability method was used to distinguish the absolute configurations of C-3 in **3** and **5**. All of **1**–**7** were tested for their chitinase inhibitory activity against *Of*Hex1 and *Of*Chi-h and cytotoxicity against A549, HGC-27 and UMUC-3 cell lines. Compounds **1**–**7** exhibited weak activity towards *Of*Hex1 and strong activity towards *Of*Chi-h at a concentration of 10.0 μM, with the inhibition rates of 0.7%–10.3% and 79.1%–95.4%, respectively. Interestingly, **1**–**7** showed low cytotoxicity against A549, HGC-27 and UMUC-3 cell lines, suggesting that good prospect of this cluster of metabolites for drug discovery.

## 1. Introduction

The biosynthetic gene clusters in one fungus species usually destine the generation of structurally versatile secondary metabolites [1]. In recent years, plenty of structurally-unique secondary metabolites have been obtained from the marine-derived fungi [2,3]. However, natural compounds from fungi usually have a high degree of chiral variety, forming one structure with different configurations, which caused a challenging task of determination for their configurations, especially when the molecules displayed high conformational flexibility [3]. Meanwhile, the stereochemistry of molecules has become one of the most important features of chiral natural products, which play a fundamental role in biology, chemistry and medicine. Thus, assigning the stereochemical characterizations attracted more and more attentions in the field of natural medicinal chemistry [4,5]. In our research on structurally-unique and biologically-active metabolites from the marine-derived fungi, the fungal strain *Penicillium polonicum* HBU-114, whose EtOAc extract exhibited the original thin-layer chromatography (TLC) and high performance liquid chromatography-under voltage (HPLC-UV) profiles of the secondary metabolites, differed from the other fungi, attracted our attention. HPLC-guided separation resulted in the isolation of three new quinazoline alkaloids, polonimides A–C (**1**–**3**) and four known analogues, aurantiomide C (**4**) [6], anacine (**5**) [6], aurantiomide A (**6**) (Appendix A) [6] and aurantiomide B (**7**) (Appendix A) [6] (Figure 1). Herein, we report the isolation, absolute configurations and bioactivity of these compounds.

## 2. Results

Polonimide A (**1**) was obtained as a yellow amorphous powder. The molecular formula of C_19_H_21_N_3_O_4_ (11 degrees of unsaturation) was established by the positive high resolution electrospray ionization mass spectroscopy (HRESIMS) data. The downfield of ^1^H NMR spectrum (Table 1, Appendix A) for **1** exhibited six proton signals, including one nitrogen-hydrogen proton signal *δ*_H_ 10.49 and five olefin proton signals (*δ*_H_ 8.13, 7.84, 7.69, 7.52 and 6.22). In addition, the ^1^H NMR spectrum of **1** showed one methoxy singlet *δ*_H_ 3.42 and two methyl doublets (*δ*_H_ 1.08 and 1.05). The ^13^C NMR spectrum (Table 2, Appendix A) of **1** revealed three carbonyl carbon signals (*δ*_C_ 159.9, 165.2 and 171.8). With careful inspection and analyses of the 1D-NMR and HSQC data (Appendix A), it was found that **1** shares the same quinazoline core as aurantiomide C (**4**), a diketopiperazine alkaloid isolated from the fungus *Penicillium aurantiogriseum* [6]. The main differences between **1** and **4** were the presence of an additional methoxy group (*δ*_H_ 3.42; *δ*_C_ 51.3) and the absence of two nitrogen-hydrogen proton signals (*δ*_H_ 7.27 and 6.73 in **4**) (Appendix A) in **1**, suggesting the presence of 17-OCH_3_ group in **1** instead of 17-NH_2_ group in **4**. The above deduction was further confirmed by the key HMBC correlation from -OCH_3_ to C-17 of **1** (Figure 2, Appendix A).

Polonimide B (**2**) was also obtained as a yellow amorphous powder with the molecular formula C_18_H_20_N_4_O_3_ (11 degrees of unsaturation) by HRESIMS spectrum. Detailed analysis of the 1D and 2D NMR spectra of **2** (Appendix A), it was found that **2** was an analogue of **1**. Combination with their NMR (Table 1 and Table 2) and HRESIMS data (Appendix A), showed that **2** differed from **1** by loss of an OCH_3_ unit (*δ*_H_ 3.42, δ_C_ 51.3 in **1**) and replaced by a NH_2_ unit (δ_H_ 7.26 and 6.71 in **2**) in **2**. Combined analysis of the differences existed in the chemical shifts of H-18, H-19, C-18 and C-19 between **2** and **4** indicated that **2** was an isomer of **4** with different geometries of the double bone C_3_=C_18_.

Polonimide C (**3**) was obtained with the molecular formula of C_18_H_22_N_4_O_3_ (10 degrees of unsaturation) by the positive HRESIMS data (Appendix A). Analysis of the ^1^H and ^13^C NMR spectra of **3** (Table 1 and Table 2, Appendix A), revealed **3** had the same anacine core structure as **2**. The main differences between **3** and **2** were the NMR chemical shifts of 2-NH, C-4, C-19 and the presence of one nitrogen-bearing methine (*δ*_H_ 4.74, *δ*_C_ 55.7 in **3**), one methylene (*δ*_H_ 2.29/1.66, *δ*_C_ 39.0 in **3**) and the absence of one olefinic quaternary carbon (C-3 in **2**) and methylene (C-18 in **2**) in **3**. It was inferred that the double bone between C-3 and C-18 in **2** was reduced in **3**, which could also be verified by the ^1^H-^1^H COSY correlations from H-3 (*δ*_H_ 4.74) to H-18 (*δ*_H_ 2.29, *δ*_H_ 1.66) and from H-18 to H-19 (δ_H_ 2.09) of **3** (Appendix A). The chemical structure of **3** was further confirmed by the key HMBC correlations from NH-2 to C-3, C-4 and C-14, from H-14 to C-1 and C-4, from H-18 to C-4 and from H-3 to C-4 of **3** (Figure 2, Appendix A). It should be noted that the chemical structure of **3** could be found in the Scifinder database with the cas registry number 154725-83-4. However, the chemical structure of cas 154725-83-4 was the same as anacine (**5**), which was originally proposed as a benzodiazepine structure by Mantle and co-workers [7] but was revised as a quinazoline structure by Sim and co-workers [8].

The geometries of the double bond C_3_=C_18_ in **1** and **2** were established with their selective 1D nuclear overhauser effect (NOE) experiments (Appendix A). In compound **1**, the irradiation of 2-NH (*δ*_H_ 10.49) resulting in no obvious enhancement of H-18 (*δ*_H_ 6.22) indicated that 2-NH and H-18 might be *trans* oriented; the downfield chemical shift of H-18 caused by the deshielding effect of 4-imine also confirmed the *Z* geometries of the double bond C_3_=C_18_. Whereas, in compound **2**, irradiation of 2-NH (*δ*_H_ 10.49) resulted in enhancement of H-18 (*δ*_H_ 5.54), was ascertained that H-18 and 2-NH in **2** were *cis* oriented, which indicated the geometries of the double bond C_3_=C_18_ was *E*.

In compound **3,** the chemical shift at C-14/15/18 (*δ*_C_ 50.7/25.6/39.0) showed some deviation from that of **5** (C-14/15/18, *δ*_C_ 53.8/29.4/47.2) (Appendix A), which suspected **3** and **5** were a pair of epimers. The position of H-3 and H-18 was in 1,4 relation, which meant it was too far to provide the (nuclear overhauser effect spectroscopy) NOESY correlations for **3** (Appendix A), thus the relative configuration between C-3 and C-14 in **3** could not be determined.

The absolute configuration of the Glutamine residue at C-14 position of **1**–**3** was established by the combination of Marfey’s method and comparing the computed electronic circular dichroism (ECD) spectra with their experimental results. The HPLC analysis showed that the Glutamine residue in compounds **1**–**3** was L-Glutamine (Figure 3). Simultaneously, the ECD curve displayed that the predicted ECD spectra of 14*S*-**1**/**2**/**3** look similar to the experimental results of **1**–**3** (Figure 4, Appendix A). Therefore, the absolute configuration of C-14 was assigned as *S* in **1**–**3**.

Analysis of the structure of **3** and the experimental ECD curves of **1**–**3** (Figure 4) revealed the absolute configuration of C-3 in **3** contributed little to its ECD Cotton effects. Since the absolute configuration at C-14 has been determined by Marfey’s analysis, the ^13^C NMR chemical shift calculation was applied to confirm the C-3 absolute configuration of **3**, with the combination of DP4plus probability method, which is one of the most sophisticated and popular strategies for chemical structure interpretation [9]. Both **3** and **5** were performed with two configurations [(3*R*,14*S*)-**3**/**5** and (3*S*,14*S*)-**3**/**5**)] for NMR calculations. The calculated NMR data of **3** and **5** were all together compared with the experimental results, respectively. The result showed that (3*R*,14*S*)-**3** was more likely than (3*S*,14*S*)-**3** (100% vs. 0%) compared with the experimental data of **3** and (3*S*,14*S*)-**5** is more likely than (3*R*,14*S*)-**5** (100% vs. 0%) compared with the experimental data of **5** (Appendix A). Subsequently, the absolute configurations of **3** and **5** were assigned as 3*R*,14*S*-**3** and 3*S*,14*S*-**5**, respectively.

The insect enzymes GH20 β-*N*-acetyl-d-hexosaminidase *Of*Hex1 and GH18 chitinase *Of*Chi-h represent important chitinolytic enzymes found in the agricultural pest Ostrinia furnacalis (Guenée) and inhibition of these enzymes have been considered a promising strategy for the development of eco-friendly pesticides. All of the isolated compounds were evaluated for their in vitro inhibitory potency against *Of*Hex1 and *Of*Chi-h, by using MU-GlcNAc and MU-(GlcNAc)_2_ as substrates, respectively. Compounds **1**–**7** exhibited weak activities towards *Of*Hex1 and strong activities towards *Of*Chi-h at a concentration of 10.0 μM (Table 3), suggesting some valuable clues regarding the structure-activity relationships. Compound **1** bearing the methoxy group on C-17 could weaken the inhibitory activities against *Of*Hex1 and *Of*Chi-h compared to **4**. Moreover, in compounds **2** and **4**, the double bond of *Z* configuration of **4** showed better activities, indicated that the geometries of the double bone C_3_=C_18_ in **2** and **4** had a direct influence on the inhibitory efficiency against *Of*Hex1 and *Of*Chi-h.

To further explore the inhibition mechanism of these quinazoline-containing diketopiperazines towards *Of*Hex1 and *Of*Chi-h, compounds **1** and **4** were firstly selected for investigating the binding mode using molecular docking to *Of*Chi-h (Figure 5a) and *Of*Hex1 (Figure 5c), respectively.

Compound **1** was found to tightly bind to the entire active pocket of *Of*Chi-h (Appendix A). Three hydrogen bonds, with bond lengths of 2.67 Å, 2.82 Å and 2.91 Å, were formed by the C-12 cycloamide-carbonyl group and C-17 ester carbonyl group of **1** with the guanidine group of ARG439 (Appendix A). The benzene ring and the lactam ring of **1** had an π-sulfur interaction with the sulfur atom of methionine MET381, with the operating distances of 4.57 Å and 5.01 Å, respectively (Appendix A). It was also found that **1** had a π-π stacking with the benzene rings of Trp268 and Phe309 (Appendix A). Alkyl hydrophobic interactions were showed between the C-18 isopropyl of **1** and Ala355/Met381 and between the 17-OCH_3_ of **1** and Val469 (Appendix A). In addition, the C-18 isopropyl of **1** bound to Tyr156 and Phe184 through mixed π/alkyl hydrophobic interactions (Appendix A).

Compound **4** was found to bind to the *Of*Hex1 in a “U” conformation (Appendix A). The C-17 carbonyl and 17-NH_2_ of **4** formed hydrogen bonds on the guanidine group of ARG220 and the carboxyl of ASP367, respectively (Appendix A). Compound **4** had a π-anion with the carboxyhydroxyl oxygen anion in the residue of GLU368 (Appendix A) and π-π stacking interaction with the indole ring of Trp490 (Appendix A). It was also found that the C-18 isopropyl group of **4** had an alkyl hydrophobic interaction with the isopropyl group of Val484 (Appendix A). Mixed π/alkyl hydrophobic interactions were also found between the C-18 isopropyl group of **4** and Trp322/Trp483 and between the benzene and lactam rings of **4** and the isopropyl group of Val327 (Appendix A).

It is noteworthy that the geometric isomers **2** and **4** had different activity value, urging to investigate the binding mode of **2** and *Of*Hex1 (Figure 5b). From the FlexibleDocking results, it was found that the π-anion between **4** and the carboxyhydroxyl oxygen anion in GLU368 was absent in **2**. Moreover, the amocarbonyl group of **4** and guanidine group of Arg220 formed two hydrogen bond interactions, while there was only one N-H hydrogen bond interaction between **2** and Arg220 (Appendix A). The above results could explain **2** was less active than **4**.

All of the isolated compounds (**1**–**7**) were also evaluated for their cytotoxic activities against human lung cancer cell line (A549), human gastric cancer cell line (HGC-27), human bladder cancer cell line (UMUC-3) and a non-tumoral cell line, human gastric epithelium (GES-1) (Appendix A). Among them, only compound **5** exhibited cytotoxicities against the three cell lines (IC_50_ = 6.0, 6.2, 7.2 μM, respectively).

## 3. Discussion

Diketopiperazines, which have been found to occur from a wide range of fungi, display a variety of bioactivities from antineoplastic, antifungal, antibacterial, to anti-inflammatory effects and have the potential to be used in the development of new drugs [10,11]. Among them, quinazoline-containing diketopiperazines, generally possess a tricyclic core of benzene−pyrimidinone−diketopiperazine, are relatively rare. Due to the configurational flexibility of the residue and little contribution to ECD Cotton effects, it was hard to assign the absolute configurations of C-3 of **3** and **5**. In the course of our study, the ^13^C NMR chemical shift calculation with the combination of DP4plus probability method was applied to assign and distinguish the C-3 absolute configuration of **3** and **5**. Furthermore, the isomers **2** and **4** and **3** and **5** showed different activity value, suggesting that the geometries of the double bone C_3_=C_18_ in **2** and **4** and the configuration at C-3 in **3** and **5**, may play an important role for bioactivities.

## 4. Experimental Section

### 4.1. General Experimental Procedures

Optical rotatory (OR) and ECD data were performed on a JASCO P-1020 and JASCO J-815 spectrometers (JASCO Corporation, Tokyo, Japan), respectively. UV and IR spectra were gathered using a Perkin-Elmer model 241 spectrophotometer (Perkin-Elmer Corporation, MA, USA) and a Nicolet NEXUS 470 spectrophotometer (Thermo Corporation, MA, USA), respectively. 1D/2D NMR spectra were measured on a Bruker AV-600 spectrometer (Thermo Corporation, Karlsruhe, Germany). HRESIMS data were recorded on a Thermo Scientific LTQ Orbitrap XL spectrometer (Thermo Corporation, MA, USA). Semipreparation HPLC (Shimadzu LC-20AT system) (Hitachi High Technologies, Tokyo, Japan) was operated using a SPD-M20A detector (Hitachi High Technologies, Tokyo, Japan) and a Waters RP-18 column (Waters Corporation, Manchester, UK). The material of Sephadex LH-20 and Silica gel used for chromatographic separation were the same as those in our previous literature [4].

### 4.2. Isolation of the Fungal Material

The fungal strain HBU-114 with the National Center for Biotechnology Information (NCBI) GenBank accession number MN623481, collected from the Bohai Sea (Huanghua, Hebei Province, China, June 2016), was identified as *Penicillium polonicum* by the molecular biological method of amplification and sequencing of the DNA sequences of the ITS region of the rRNA gene. It was deposited in the College of Pharmaceutical Sciences, Hebei University. The fungus HBU-114 was cultivated in rice medium (80 g rice, 60 mL H_2_O, 2.0 g sea salt in each Erlenmeyer flask) in a total of forty Erlenmeyer flasks at 28 °C for 28 days. Mixture of CH_2_Cl_2_/MeOH (1:1, *v*:*v*) was used to extract the fermented rice substrate for six times. The organic extract was evaporated to remove solvent, after which it was extracted with EtOAc and H_2_O (1:1, *v*:*v*) for six times and evaporated to dryness to give the EtOAc extract (10.8 g). The extract was separated by silica gel column chromatography (CC) with EtOAc-petroleum ether (PE) (0–100% EtOAc) to give six fractions (Fr.1–Fr.6). Fr.5, eluted with 80% EtOAc–PE (4:1, *v*:*v*), was applied to a Sephadex LH-20 CC (CH_2_Cl_2_/MeOH (1:1, *v*:*v*)) to remove the pigment to give Fr.5-1–Fr.5-2. Then, Fr.5-2 was further separated by silica gel CC using mixtures of CH_2_Cl_2_ and MeOH (20:1, *v*:*v*) to offer Fr.5-2-1–Fr.5-2-4. Among them, Fr.5-2-2 was purified by ODS column eluted with 80% MeOH/H_2_O and then separated by HPLC on a waters RP-18 column (XBridge OBD, 5 μm, 10 × 250 mm, MeOH-H_2_O (40:60, *v*:*v*) to obtain polonimide A (**1**) (6.5 mg, 51.0 min), polonimide B (**2**) (5.6 mg, 36.0 min) and **7** (8.9 mg, 24.5 min), respectively. Fr.5-2-3 was chromatographed on silica gel eluting with CH_2_Cl_2_–MeOH (30:1) and further purified by HPLC on a waters RP-18 column (XBridge OBD, 5 μm, 10 × 250 mm, MeOH-H_2_O (40:60, *v*:*v*) to give polonimide C (**3**) (8.5 mg, 33.0 min), **4** (355.2 mg, 21.0 min), **5** (3.4 mg, 17.5 min) and **6** (8.2 mg, 37.5 min), respectively.

*Polonimide A (**1**):* amorphous powder; ${\left[\alpha \right]}_{D}^{20}$ +14.0 (*c* 0.3, CH_3_OH); UV (MeOH) λ_max_ (log *ε*) 245 (1.50) nm; ECD (MeOH) λ_max_ (Δ*ε*) 222 (+3.4), 245 (+1.1) nm; IR (KBr) *v*_max_ 3198, 1636, 1602, 1588, 1563, 761 cm^−1^; ^1^H and ^13^C NMR, see Table 1 and Table 2; HRESIMS *m*/*z* 378.1424 [M + Na]^+^ (calcd for C_19_H_21_N_3_O_4_Na, 378.1424 [M + Na]^+^).

*Polonimide B (**2**):* amorphous powder; ${\left[\alpha \right]}_{D}^{20}$ +24.0 (*c* 0.3, CH_3_OH); UV (MeOH) λ_max_ (log *ε*) 350 (1.50) nm; ECD (MeOH) λ_max_ (Δ*ε*) 222 (+3.4), 315 (+1.0) nm; IR (KBr) *v*_max_ 3191, 1634, 1601, 1579, 1566, 765 cm^−1^; ^1^H and ^13^C NMR, see Table 1 and Table 2; HRESIMS *m/z* 378.1408 [M + Na]^+^ (calcd for C_18_H_20_N_4_O_3_Na, 378.1433 [M + Na]^+^).

*Polonimide C (**3**):* amorphous powder; ${\left[\alpha \right]}_{D}^{20}$ +171.0 (*c* 0.3, CH_3_OH); UV (MeOH) λ_max_ (log *ε*) 330 (1.50) nm; ECD (MeOH) λ_max_ (Δ*ε*) 210 (-0.7), 228 (1.7) nm; (KBr) *v*_max_ 3193, 1633, 1603, 773 cm^−1^; ^1^H and ^13^C NMR, see Table 1 and Table 2; HRESIMS *m*/*z* 365.1568 [M + Na]^+^ (calculated for C_18_H_22_N_4_O_3_Na, 365.1590 [M + Na]^+^).

### 4.3. General Computational Procedure

Quantum chemical calculations for **1**–**3** and **5** were carried out on the basis of previous references (gas phase) [12,13,14]. Chemical structures of **1**–**3** and **5** were constructed and used for conformational searches using MMFF94S force field by the BARISTA 7.0 software (CONFLEX Corporation Tokyo, Japan). Of all the geometries, those with relative energy of 0–10.0 kcal/mol (93 stable conformers for **1**, 85 stable conformers for **2**, 49 stable conformers for **3** and 58 stable conformers for **5**) were optimized at the B3LYP/6-311+G(d) level, then those with a relative energy of 0–2.5 kcal/mol (15 conformers for **1**, 27 conformers for **2**, 9 conformers for **3** and 16 conformers for **5**) were chosen for ECD calculations at the B3LYP/6-311++G(2d,p) level and simulated using SpecDis 1.71 [15]. In addition, DP4plus applications were used to assign the absolute configurations of **3** and **5**, the optimized conformers were calculated at the B3LYP/6-311+G(d,p)//B3LYP/6-311+G(d,p) level for the unshielded tensor values. All of the calculations were processed with Gaussian 09 package [16].

### 4.4. Preparation and Analysis of Marfey’s Derivatives

Compounds **1**–**3** (0.2 mg, respectively), dissolved in 0.5 mL of 6N HCl under the temperature of 110 °C, were hydrolyzed for 4 h. After temperature of the solutions dropped to 25 °C, the mixture were evaporated to dryness under vacuum with addition of distilled H_2_O to remove the trace HCl, then redissolved it in H_2_O (50 μL). The divided hydrolysate were treated with 200 μL of 0.5% (*w*/*v*) 1-fluoro-2-4-dinitrophenyl-5-L-alanine amide (FDAA) in acetone and 20 μL 1N NaHCO_3_ in order of precedence. The mixture was stirred at 45 °C for 40 min and then it was quenched through the addition of 20 μL of 2N HCl. The mixture was evaporated to give the resulting residues, which was dissolved in MeOH (20 μL) [17]. Similarly, the standard amino acid _L_-Glutamine and _D_-Glutamine were derivatized with FDAA using the same procedure as that of **1**–**3**. And the derivatives were analyzed by HPLC with linear isocratic elution (MeOH-H_2_O (70:30, *v*:*v*)) detected at 254 nm.

### 4.5. Enzymes Inhibitory Activity Assay

In a final assay volume of 100 μL, enzyme was incubated with substrate (20 μM MU-(GlcNAc)_2_ for *Of*Chi-h and 50μM MU-GlcNAc for *Of*Hex1) in 20 mM sodium phosphate buffer (pH 6.0 for *Of*Chi-h, pH 6.5 for *Of*Hex1) containing 10 μM inhibitor at 30 °C. The reaction in the absence of inhibitor was used as a control. After reacting for an appropriate time (30min), an equal volume of 0.5 M Na_2_CO_3_ was added to the reaction mixture to terminate the reaction and the fluorescence of the liberated MU was quantitated using a Varioskan Flash microplate reader, with excitation and emission wavelengths of 360 and 450 nm [18].

### 4.6. Molecular Docking

The complex crystal structure of *Of*Hex1-PUGNAc (Protein Data Bank (PDB) entry code: 3OZP) [19] or *Of*Chi-h-chitohepatose (PDB entry code: 5GQB) [20] was used as the starting model for molecular docking employing the Discovery Studio 2017 software. Before docking calculations, conformational searches of the compounds **1** and **4** were performed with the GMMX conformer calculation (Force field: MMFF94, energy window: 5.0 kcal/mol) in GaussView 6.0. Then, the top 5 conformations with the lowest energy were optimized at the B3LYP/6-31G(d) basis set using density functional theory (DFT) in Gaussian 16. To simulate real conditions, the solvent effects of H_2_O were studied using the solvation model based on density (SMD). For the protein, the protein preparation processes were undergone, such as removing water molecules, adding hydrogen atoms and supplementing amino acid residues. The Flexible Docking protocol, which allows for some receptor flexibility during docking of flexible ligands [21] that employs CHARMm in Discovery Studio 2017 software, was used in this study. The receptor binding sites were determined from the PDB site records.

### 4.7. Cytotoxic Assay

All of the isolated compounds **1**–**7** were evaluated for cytotoxic activity in vitro according to MTT method [22]. Three human tumor cell lines were included—human lung cancer cell line (A549), human gastric cancer cell line (HGC-27), human bladder cancer cell line (UMUC-3) and a non-tumoral cell line, human gastric epithelium (GES-1). The positive control was cisplatinum (DDP).

Cell lines and cell culture Human A549 and HGC-27 cancer cells were obtained from Chinese Academy of Medical Sciences, Basic Medicine Cell Center (Beijing, China). Human UMUC-3 cancer cells were obtained from Cell Resource Center, Shanghai Institute of Life Sciences, Chinese Academy of Sciences (Shanghai, China). Cells were cultured in RPMI-1640 media supplemented with 10% fetal bovine serum (FBS), 100 U/mL penicillin, and 100 mg/mL streptomycin at 37 °C in a 5% CO_2_ atmosphere.

## 5. Conclusions

Three new quinazoline alkaloids polonimides A–C (**1**–**3**), were obtained from the marine-derived fungus *Penicillium polonicum*. The relative and absolute configurations of **1**–**3** were comprehensively determined by combination of 1D NOE experiments, modified Marfey’s analysis, ECD and NMR chemical shift calculations. The quinazoline-containing diketopiperazines (**1**–**7**) with low cytotoxicity but potent chitinase inhibitory activity also indicated good prospect of this cluster of metabolites for drug discovery.

## Figures and Tables

**Figure 1 marinedrugs-18-00479-f001:**
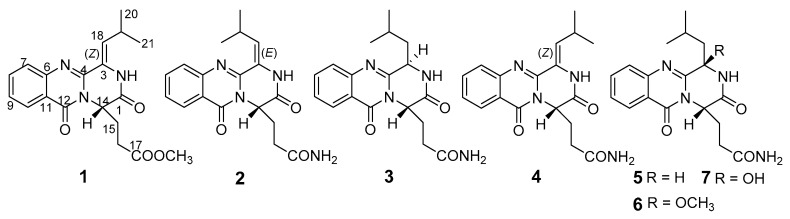
Chemical structures of **1**–**7**.

**Figure 2 marinedrugs-18-00479-f002:**
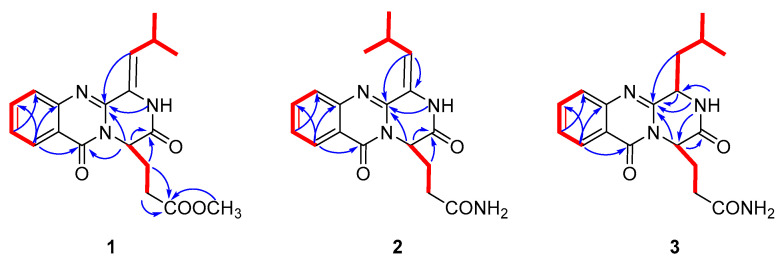
^1^H-^1^H correlation spectroscopy (COSY) (Appendix A) (bold) and Key heteronuclear multiple bond correlation (HMBC) (arrows) of **1**–**3**.

**Figure 3 marinedrugs-18-00479-f003:**
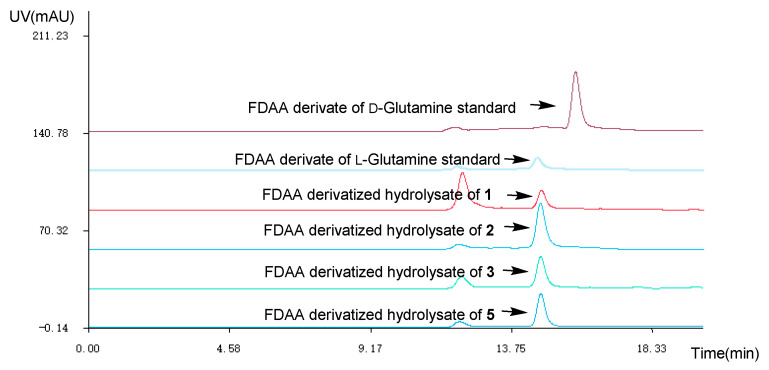
HPLC at 254 nm of the Marfey’s analysis (1-fluoro-2-4-dinitrophenyl-5-L-alanine amide (FDAA) derivate of _L_-Glutamine standard *t*_R_ 14.5 min; FDAA derivate of _D_-Glutamine standard *t*_R_ 15.8 min; MeOH-H_2_O (70:30, *v*:*v*), *v* = 2.0 mL/min).

**Figure 4 marinedrugs-18-00479-f004:**
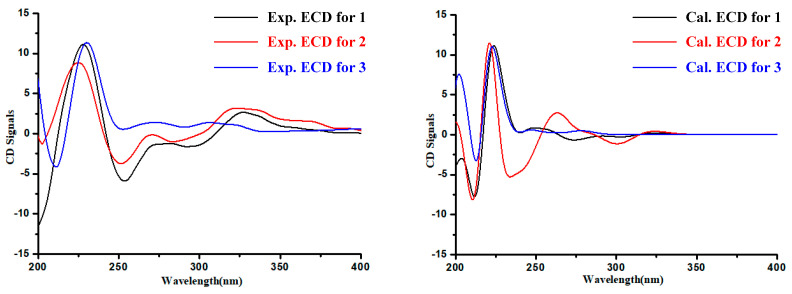
Experimental and calculated electronic circular dichroism (ECD) spectra of **1**–**3**.

**Figure 5 marinedrugs-18-00479-f005:**
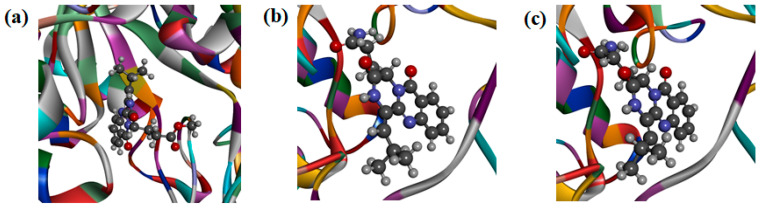
(**a**) Docking simulations of **1** with target protein *Of*Chi-h. (**b**) Docking simulations of **2** with target protein *Of*Hex1. (**c**) Docking simulations of **4** with target protein *Of*Hex1.

**Table 1 marinedrugs-18-00479-t001:** ^1^H NMR Data (*δ*) of 1–3 and 5 (600 MHz, DMSO-*d_6_*, *J* in Hz).

No.	1	2	3	5
2	10.49, brs	10.49, brs	8.53, brs	8.91, d (4.2)
3	-	-	4.74, dd (7.8, 3.6)	4,40–4.43, m
7	7.69, d (7.8)	7.65, d (8.4)	7.68, d (8.4)	7.66, d (7.8)
8	7.84, dd (7.8, 7.2)	7.86, dd (8.4, 7.2)	7.85, dd (8.4, 7.2)	7.84, dd (7.8, 7.2)
9	7.52, dd (7.8, 7.2)	7.57, dd (7.8, 7.2)	7.56, dd (7.8, 7.2)	7.54, dd (7.8, 7.2)
10	8.13, d (7.8)	8.16, d (7.8)	8.15, d (7.8)	8.15, d (7.8)
14	5.19, dd (6.6, 6.0)	5.12, dd (6.6, 6.0)	5.09, dd (7.8, 6.6)	4.86, dd (9.0, 6.0)
15	2.12–2.17, m	2.02–2.06, m	2.17–2.20, m	2.09–2.14, m
-	2.02–2.07, m	-	2.12–2.14, m	2.01–2.06, m
16	2.38–2.43, m	2.11–2.14, m	2.21–2.26, m	2.30–2.35, m
-	2.32–2.37, m	-	-	2.38–2.44, m
18	6.22, d (10.8)	5.54, d (9.6)	2.28–2.31, m	1.74–1.81, m
-	-	-	1.64–1.68, m	-
19	2.94–3.00, m	3.74–3.79, m	2.07–2.11, m	1.87–1.93, m
20	1.05, d (6.6)	1.02, d (6.6)	0.97, d (6.6)	0.97, d (6.6)
21	1.08, d (6.6)	1.20, d (6.6)	0.98, d (6.6)	0.99, d (6.6)
17-OCH_3_	3.42, s, 3H	-	-	-
17-NH_2_	-	7.26, s	7.29, s	7.36, s
-	-	6.71, s	6.75, s	6.78, s

**Table 2 marinedrugs-18-00479-t002:** ^13^C NMR Data (*δ*) of 1–3 and 5 (150 MHz, DMSO-*d_6_*).

No.	1	2	3	5
1	165.2, C	165.5, C	167.9, C	166.6, C
3	126.6, C	124.8, C	55.7, CH	54.9, CH
4	145.5, C	144.8, C	152.2, C	152.0, C
6	147.0, C	146.7, C	146.6, C	147.0, C
7	126.3, CH	127.5, CH	127.3, CH	126.7, CH
8	134.7, CH	134.7, CH	134.7, CH	134.7, CH
9	126.7, CH	127.2, CH	127.0, CH	126.7, CH
10	125.3, CH	126.3, CH	126.3, CH	126.2, CH
11	119.7, C	119.7, C	119.8, C	119.7, C
12	159.9, C	159.7, C	160.1, C	160.1, C
14	54.3, CH	54.7, CH	50.7, CH	53.8, CH
15	27.2, CH_2_	28.0, CH_2_	25.6, CH_2_	29.4, CH_2_
16	29.2, CH_2_	30.9, CH_2_	31.3, CH_2_	32.2, CH_2_
17	171.8, C	172.4, C	172.7, C	172.8, C
18	127.1, CH	131.0, CH	39.0 *^a^*, CH_2_	47.2, CH_2_
19	25.0, CH	26.5, CH	23.8, CH	24.0, CH
20	22.0, CH_3_	23.1, CH_3_	23.3, CH_3_	23.0, CH_3_
21	22.3, CH_3_	22.4, CH_3_	21.7, CH_3_	21.4, CH_3_
17-OCH_3_	51.3, CH_3_	-	-	-

*^a^* which was speculated in the HSQC spectrum according to the correlations from *δ*_H_ 2.29/1.66 to *δ*_C_ 39.0.

**Table 3 marinedrugs-18-00479-t003:** Chitinase Inhibitory Activity for Compounds **1**–**7**.

Compounds	Inhibition Rate (%)			
	*Of*Hex1	*Of*Chi-h		
	10.0 μM	10.0 μM	1.0 μM	0.2 μM
**1**	0.7	91.9	75.1	28.1
**2**	3.8	79.1	74.3	4.0
**3**	1.4	86.1	73.1	15.8
**4**	10.3	95.4	85.5	23.2
**5**	6.8	92.3	75.9	20.1
**6**	7.4	90.5	83.9	3.2
**7**	5.9	85.7	77.6	21.7

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
