# Peer review of "Absolute Configurations and Chitinase Inhibitions of Quinazoline-Containing Diketopiperazines from the Marine-Derived Fungus *Penicillium polonicum"

_marinedrugs, 2020, doi:10.3390/md18090479_

Round 1

Reviewer 1 Report

Review of the article: "Absolute Configurations and Chitinase Inhibitions of Quinazoline-Containing Diketopiperazines from the Marine-Derived Fungus Penicillium polonicum

Submission ID - marinedrugs-913955

The manuscript is very interesting and well prepared. I have only several (minor of importance) critical comments. However, the most important drawback of the manuscript is the fact that the discussion is very weak in fact the authors did not present any discussion. In my opinion the manuscript can be presented but short discussion should be added.

Detailed comments:

Abstract – I would be grateful if the authors could consider presenting some more details regarding results of molecular docking, chitinase inhibitory activity (the most important values of inhibition rates) and cytotoxicity assay.

Introduction

Well prepared – no critical comments.   

Materials and methods

Line 184 – please provide some more details – which “molecular biology methods” were used.

Lines 192-207 – In my opinion this fragment could be improved. It would be necessary to write it clearly which fractions contained Polonimide A, Polonimide B and Polonimide C respectively.

Enzyme inhibitory activity  assay – I accept this methodology. However, it would be better if the authors determined the concentrations of the agents that caused 50% of inhibition of the enzyme activity.

Lines 249-252 – The cytotoxicity assay should be also performed for healthy cell lines

Results

Results are very interesting and well presented. I have only one comment. As I wrote above - it would be better if the authors determined the concentrations of the agents that caused 50% of inhibition of the enzyme activity.

 Discussion  - in fact this section does not exists. It must be  completed – even short section would be required. The authors should discuss their results with the results presented by other authors, they also should present/consider some propositions of application of these agents.  

Final decision – major revision.

Author Response

Dear Reviewer,

Thank you very much for your pertinent comments on our paper. We have revised our manuscript carefully according to your comments. Our responds to questions are as follows. Any other questions, please contact us. Thanks again for your effort for our papers.

With Best Regards.

Sincerely Yours

Fei Cao

Editor’s Comments:

The manuscript is very interesting and well prepared. I have only several (minor of importance) critical comments. However, the most important drawback of the manuscript is the fact that the discussion is very weak in fact the authors did not present any discussion. In my opinion the manuscript can be presented but short discussion should be added.

Q1: Abstract – I would be grateful if the authors could consider presenting some more details regarding results of molecular docking, chitinase inhibitory activity (the most important values of inhibition rates) and cytotoxicity assay.

A1: Thank you very much for your suggestion. We have added: “Compounds 17 exhibited weak activites towards OfHex1 and strong activites towards OfChi-h at a concentration of 10.0 μM, with the inhibition rates of 0.7−10.3% and 79.1−95.4%, respectively. Interestingly, 17 showed low cytotoxicity against A549, HGC-27, and UMUC-3 cell lines, suggesting that good prospect of this cluster of metabolites for drug discovery” in the revised manuscript.

Q2: Materials and methods – Line 184 – please provide some more details – which “molecular biology methods” were used.

A2: Thanks very much for your effort for our papers. We have revised it as: molecular biological method of amplification and sequencing of the DNA sequences of the ITS region of the rRNA gene.

Q3: Materials and methods – Lines 192-207 – In my opinion this fragment could be improved. It would be necessary to write it clearly which fractions contained Polonimide A, Polonimide B and Polonimide C respectively.

A3: Thanks a lot for your kind advice. More detail of isolation of Polonimide A, Polonimide B and Polonimide C was revised as: Fr.5-2 was further separated by silica gel CC using mixtures of CH2Cl2 and MeOH (20:1, v:v) to offer Fr.5-2-1–Fr.5-2-4. Among them, Fr.5-2-2 was purified by ODS column eluted with 80% MeOH/H2O, and then separated by HPLC on a waters RP-18 column (XBridge OBD, 5 μm, 10 × 250 mm, MeOH-H2O (40:60, v:v) to obtain polonimide A (1) (6.5 mg, 51.0 min), polonimide B (2) (5.6 mg, 36.0 min), and 7 (8.9 mg, 24.5 min), respectively. Fr.5-2-3 was chromatographed on silica gel eluting with CH2Cl2–MeOH (30:1), and further purified by HPLC on a waters RP-18 column (XBridge OBD, 5 μm, 10 × 250 mm, MeOH-H2O (40:60, v:v) to give polonimide C (3) (8.5 mg, 33.0 min), 4 (355.2 mg, 21.0 min), 5 (3.4 mg, 17.5 min), and 6 (8.2 mg, 37.5 min), respectively.

Q4: Enzyme inhibitory activity assay – I accept this methodology. However, it would be better if the authors determined the concentrations of the agents that caused 50% of inhibition of the enzyme activity.

A4: Thanks a lot for your suggestion. Indeed, it was better to report the 50% of inhibition of the enzyme activity. However, in the previous experiments, the amount of compounds had been used up. Thus, we could not obtain these data.

Q5: Lines 249-252 – The cytotoxicity assay should be also performed for healthy cell lines.

A5: Thanks for your advice. A non tumoral cell line, human gastric epithelium (GES-1), was also used to evaluate for their toxicities of 17. The results showed that compounds 17 exhibited no toxicity.

Q6: Results are very interesting and well presented. I have only one comment. As I wrote above - it would be better if the authors determined the concentrations of the agents that caused 50% of inhibition of the enzyme activity.

A6: Thank you again for your pertinent comments on our paper. Indeed, it was better to report the 50% of inhibition of the enzyme activity. However, in the previous experiments, the amount of compounds had been used up. Thus, we could not obtained these data.

Q7: Discussion - in fact this section does not exists. It must be completed – even short section would be required. The authors should discuss their results with the results presented by other authors, they also should present/consider some propositions of application of these agents.

A7: Thank you for your effort on our paper. We have added the “Discussion Section” in the revised manuscript.

Reviewer 2 Report

The authors identified three new quinazoline containing diketopiperazines. The structures of new compounds are very similar to anacine and aurantiomides. However, their chitinase inhibitory activities are not reported before. Therefore, it would be of interest to the readers of Mar. Drugs. In the manuscript, there are several issues to be clarified. Therefore, it would be considered before further consideration for acceptance.

  1. Please unify compound names. New compounds are described aurantiomides in the conclusion paragraph.
  2. Bond length and angle of compounds in Figure 1 looks strange. Please re-draw structures.
  3. Please add more KEY HMBC correlations to support tri-cyclic core structures in Figure 2.
  4. The authors used reference number throughout the manuscript. However, the number is not found in the reference list.
  5. The authors described the compound 5 was reported in the reference 7. However, the structure was not found in the reference. Please describe the history on the structure and the compound.
  6. The authors determined the geometry of 1 and 2 based on 1D NOE data and chemical shift speculation. However, selective 1D NOE data of compound 1 was not included in Supplementary Materials (SM). Additionally, the chemical shift indication on Figures S12 looks different from the table and manuscript. Please confirm the chemical shifts in Figure S12.
  7. Line 96: The authors indicated H-3 and H-18 are in 1,4 relation. Please confirm the number and correct it.
  8. The authors provided HPLC chromatograms of FDAA derivatives of standards and 1-3 hydrolysates. In order to compare the structure of compound 5, it is recommended to add the chromatogram of FDAA derivatives of compound 5 hydrolysates in Figure 3.
  9. Quinazoline structure seems to be resistance against acidic condition, and glutamic acid derived from glutamine in quinazoline attached diketopiperazines will not be easily released. However, the authors described general acid hydrolysis procedure in the experimental. Please provide any literature for the cleavage of quinazolines or describe more detail for the release of glutamic acid unit from compounds 1-3.
  10. The authors used DP4+ calculation for absolute configuration ananlysis of 3 and 5. In this approach, conformer study will be very critical. However, the authors did not include the detail for the conformer study. Please add software information (name and version) and selected conformer numbers of each compound. Also, the authors provide the coordinate only for the lowest-energy conformer of each compound in SM. It would be proper to include those data of all the conformers used in this study.

Simple Typos

Line 82: sam -> same

Line 98, 100, 101, 224, Figure 3 and its legend : Glutamine -> glutamine

Line 216: marfey’s -> Marfey’s

Please add a space between “number” and “unit”.

Line 217: 6N -> 6 N

Line 221: 1N -> 1 N

Line 222: 2N -> 2 N

Line 229: 50uM > 50 uM

Author Response

Dear Reviewer,

Thanks very much for your comments. We have revised our manuscript carefully according to your comments. The changes in our new version of manuscript were highlighted in red. Any other questions, please contact us. Thanks again for your effort for our papers.

With Best Regards.

Sincerely Yours

Fei Cao

Editor’s Comments:

Comments and Suggestions for Authors. The authors identified three new quinazoline containing diketopiperazines. The structures of new compounds are very similar to anacine and aurantiomides. However, their chitinase inhibitory activities are not reported before. Therefore, it would be of interest to the readers of Mar. Drugs. In the manuscript, there are several issues to be clarified. Therefore, it would be considered before further consideration for acceptance.

Q1: Please unify compound names. New compounds are described aurantiomides in the conclusion paragraph.

A1: Thank you very much for your suggestion. We have revised it according your advice.

Q2: Bond length and angle of compounds in Figure 1 looks strange. Please re-draw structures.

A2: Thanks very much for your effort for our papers. We have re-drawn Figure 1 in the revised manuscript.

Q3: Please add more KEY HMBC correlations to support tri-cyclic core structures in Figure 2.

A3: Thanks a lot for your kind advice. We have added more Key HMBC correlations in Figure 2.

Q4: The authors used reference number throughout the manuscript. However, the number is not found in the reference list.

A4: Thanks very much. We have added the number of the reference list in the revised manuscript.

Q5: The authors described the compound 5 was reported in the reference 7. However, the structure was not found in the reference. Please describe the history on the structure and the compound.

A5: Thanks very much for your kind advice. We have revised “However, the the chemcial structure of cas 154725-83-4 was the sam as anacine (5) in the corresponding literature [7].” as “However, the the chemcial structure of cas 154725-83-4 was the same as anacine (5), which was originally proposed as a benzodiazepine structure by Mantle and co-workers [7], but was revised as a quinazoline structure by Sim and co-workers [8].” in the revised manuscript.

Q6: The authors determined the geometry of 1 and 2 based on 1D NOE data and chemical shift speculation. However, selective 1D NOE data of compound 1 was not included in Supplementary Materials (SM). Additionally, the chemical shift indication on Figures S12 looks different from the table and manuscript. Please confirm the chemical shifts in Figure S12.

A6: Thanks for your kind advice. We have added the 1D NOE data of 1 in the Figure S12. Also, we have re-adjusted the chemical shift indication of compound 2 in the Figure S12.

Q7: The authors provided HPLC chromatograms of FDAA derivatives of standards and 1-3 hydrolysates. In order to compare the structure of compound 5, it is recommended to add the chromatogram of FDAA derivatives of compound 5 hydrolysates in Figure 3.

A7: Thanks a lot for your suggestion. We have added the chromatogram of FDAA derivatives of compound 5 hydrolysates in Figure 3.

Q8: Quinazoline structure seems to be resistance against acidic condition, and glutamic acid derived from glutamine in quinazoline attached diketopiperazines will not be easily released. However, the authors described general acid hydrolysis procedure in the experimental. Please provide any literature for the cleavage of quinazolines or describe more detail for the release of glutamic acid unit from compounds 1-3.

A8: Thank you for your professional comments. Indeed, from the chemical structure, quinazoline structure seems to be resistance against acidic condition. However, it could be carried out. For another examples, Mar. Drugs 2019, 17, 250; doi:10.3390/md17050250, dx.doi.org/10.1021/np100880b J. Nat. Prod. 2011, 74, 1284–1287, Natural Product Communications Vol. 8 (8) 2013, 8(8), 1071-1074, and so on, could provide additional evidences.

A9: The authors used DP4+ calculation for absolute configuration ananlysis of 3 and 5. In this approach, conformer study will be very critical. However, the authors did not include the detail for the conformer study. Please add software information (name and version) and selected conformer numbers of each compound. Also, the authors provide the coordinate only for the lowest-energy conformer of each compound in SM. It would be proper to include those data of all the conformers used in this study.

A9: Thanks very much for your effort for our papers. We have added software information (name and version) and selected conformer numbers of each compound in “4.3. General computational procedure section”, and added the all the coordinate of the conformers in the SM.

Round 2

Reviewer 1 Report

The authors have taken into account all my suggestions. In my opinion the manuscript can be accepted in current form. 

Reviewer 2 Report

The authors revised manuscript as suggested by reviewer. Therefore, the manuscript could be published to readers of Mar. Drugs.